# Malnutrition Increases Hospital Length of Stay and Mortality among Adult Inpatients with COVID-19

**DOI:** 10.3390/nu14061310

**Published:** 2022-03-21

**Authors:** Tyrus Vong, Lisa R. Yanek, Lin Wang, Huimin Yu, Christopher Fan, Elinor Zhou, Sun Jung Oh, Daniel Szvarca, Ahyoung Kim, James J. Potter, Gerard E. Mullin

**Affiliations:** 1Department of Medicine, Division of Gastroenterology and Hepatology, Johns Hopkins University School of Medicine, Baltimore, MD 21205, USA; tvong2@jhmi.edu (T.V.); lryanek@jhmi.edu (L.R.Y.); hyu20@jhmi.edu (H.Y.); chris.fan@jhmi.edu (C.F.); zhou.elinor@gmail.com (E.Z.); soh51@jhmi.edu (S.J.O.); dszvarca1@jhmi.edu (D.S.); ahyoung.j.kim@gmail.com (A.K.); jpotter@jhmi.edu (J.J.P.); 2Department of Epidemiology, Johns Hopkins Bloomberg School of Public Health, Baltimore, MD 21205, USA; linwang@jhu.edu

**Keywords:** COVID-19, SARS-CoV-2, nutrition, malnutrition, vitamin D, mortality

## Abstract

Background: Malnutrition has been linked to adverse health economic outcomes. There is a paucity of data on malnutrition in patients admitted with COVID-19. Methods: This is a retrospective cohort study consisting of 4311 COVID-19 adult (18 years and older) inpatients at 5 Johns Hopkins-affiliated hospitals between 1 March and 3 December 2020. Malnourishment was identified using the malnutrition universal screening tool (MUST), then confirmed by registered dietitians. Statistics were conducted with SAS v9.4 (Cary, NC, USA) software to examine the effect of malnutrition on mortality and hospital length of stay among COVID-19 inpatient encounters, while accounting for possible covariates in regression analysis predicting mortality or the log-transformed length of stay. Results: COVID-19 patients who were older, male, or had lower BMIs had a higher likelihood of mortality. Patients with malnutrition were 76% more likely to have mortality (*p* < 0.001) and to have a 105% longer hospital length of stay (*p* < 0.001). Overall, 12.9% (555/4311) of adult COVID-19 patients were diagnosed with malnutrition and were associated with an 87.9% increase in hospital length of stay (*p* < 0.001). Conclusions: In a cohort of COVID-19 adult inpatients, malnutrition was associated with a higher likelihood of mortality and increased hospital length of stay.

## 1. Introduction

Coronavirus disease of 2019 (COVID-19) is caused by the severe acute respiratory syndrome coronavirus 2 (SARS-CoV-2) and is responsible for the third major coronavirus (CoV) pandemic in recent history [1]. COVID-19 remains a highly contagious infectious disease that can rapidly escalate to respiratory failure and death [2]. Overall, most adults infected by COVID-19 self-resolve, while approximately 20% progress and many require hospitalization. The elderly and those with comorbidities (immunocompromise, cardiopulmonary disease, cancer, diabetes, obesity, and kidney failure) are at heightened risk of experiencing rapid disease progression and mortality [3]. These comorbidities have in common chronic systemic inflammation, a higher prevalence of sarcopenia, loss of lean body mass, and protein-calorie malnutrition, which impairs immunity to infectious agents, such as SARS-CoV-2 [4,5,6,7,8]. The COVID-19 pandemic has heightened the risks of sarcopenia and protein-calorie malnutrition due to restrictions in physical activity from lockdowns, inaccessibility to exercise facilities, limited public transportation, and food insecurity from supply chain disruptions [9]. Symptomatic infection by SARS-CoV-2 results in a catabolic inflammatory response coupled with inherent physical immobility, poor nutrient intake from dysgeusia and anosmia, and frequent gastrointestinal involvement [10]. Collectively, acute symptomatic SARS-CoV-2 infections deplete lean body mass and vital micronutrients, which may further impair immunity and heighten COVID-19 disease severity. Malnutrition in the hospital setting is a known risk factor for adverse health economic outcomes [11]. Early interventions for malnutrition in hospitalized patients with chronic obstructive pulmonary disease and heart failure has improved health economic outcomes, such as length of hospital stay (LOS) and mortality [12,13]. Our study aims to determine the prevalence of malnutrition in adult inpatients admitted for SARS-CoV-2 disease and whether malnutrition is an independent risk factor for prolonged length of hospital stay and mortality.

## 2. Materials and Methods

### 2.1. Study Design and Participants

We conducted a retrospective cohort study of 4311 COVID-19 adult (18 years and older) inpatients at Johns Hopkins Medicine, which contains 5 affiliated hospitals: Johns Hopkins Hospital, Baltimore, Maryland; Bayview Hospital, Baltimore, Maryland; Howard County General Hospital, Columbia, Maryland; Suburban Hospital, Bethesda, Maryland; and Sibley Hospital, Bethesda, Maryland. Patient information was collected between 1 March 2020 and 3 December 2020. The study protocol of the JH-CROWN database was reviewed by CADRE (COVID-19 Data Research Evaluation) and the CCDA (Core for Clinical Research Data Acquisition). The institutional review board approved this study under IRB00253531 on 24 July 2020, and approved the usage of secondary research for COVID-19-related protocols. Patient data were derived from JH-CROWN, the COVID-19 Precision Medicine Analytics Platform (PMAP) Registry (PMID 32960645). Initial data extraction was provided by the Institute for Clinical and Translational Research (ICTR) and reviewed by the COVID and Data Research Evaluation Committee (CADRE).

Patients diagnosed with nasopharyngeal COVID-19 mRNA by polymerase chain reaction testing were identified by using the ICD-10 billing code of U07.1. Malnourishment among patients was determined by registered nurses by means of the malnutrition universal screening tool (MUST) [14], then confirmed by registered dietitians using global leadership initiative on malnutrition (GLIM) guidelines [15]. Malnutrition was identified in the database by the World Health Organization’s International Classification of Diseases 10th revision (ICD-10) billing code of E43 or E46. Data were extracted from the provided extraction using Python 3, version 3.7.5, kernel in JuypterLab, version 1.1.4. Extracted patient data consist of demographics, comorbidities, symptoms, vitals, O2 device events, medications, laboratory results, and medical history. Initial admission as an inpatient was used as time zero for all analyses.

The parameters of interest included those commonly observed in the hospital setting that includes demographics, such as sex, age, race, and body mass index (BMI; kg/m^2^). The hospital records denoted whether the patient encounter was an admission to the inpatient or the intensive care unit (ICU) at the time of the encounter. Variables measured in the ICU included ventilator duration, oxygen device events, vitals, and the World Health Organization score (WHO score) for determining COVID-19 severity in ICU patients. The WHO score ranged from 0 to 8 and was determined when the data were extracted. The classifications for each score are the following: 0 = No clinical or virological evidence of infection; 1 = No limitation of activities; 2 = Limitation of activities; 3 = Hospitalized, no oxygen therapy; 4 = Oxygen by mask or nasal prongs; 5 = Non-invasive ventilation or high-flow oxygen; 6 = Intubation and mechanical ventilation; 7 = Ventilation + additional organ support − pressors, renal replacement therapy (RRT), extracorporeal membrane oxygenation (ECMO); and 8 = Death [15]. Peak WHO scores during the admission were used in this analysis. Scores of 3–5 were combined to indicate mild or moderate severity, and scores of 6–7 were combined to indicate severe COVID-19. Patient comorbidities were derived from reported ICD-10 billing codes indicated by hospital staff. In-hospital mortality was assessed using the statistical analysis plan detailed below.

### 2.2. Statistical Analysis

#### 2.2.1. Descriptive Analysis

Data are expressed as the frequencies and percentages for categorical variables, means and standard deviations, or medians and interquartile ranges for continuous variables. Chi-squared tests were used to compare categorical variables between groups (malnourished or not malnourished; alive or deceased), and Student’s *t* or Wilcoxon rank-sum tests were used to compare continuous variables between groups (malnourished or not malnourished; alive or deceased).

#### 2.2.2. Multivariate Analysis

Multivariable logistic regression analysis was used to predict mortality and to estimate the effect of malnutrition controlling for age, sex, race, body mass index, diabetes, hypertension, diarrhea, COPD, and admission source. Linear regression analysis was used to predict the log-transformed length of hospital stay and to estimate the effect of malnutrition controlling for age, sex, race, body mass index, diabetes, hypertension, diarrhea, COPD, and admission source in the full sample set. The WHO severity score was added to the model for those hospitalized in the ICU; a sensitivity analysis analyzed only those who survived to hospital discharge. Missing variables were assumed to be missing at random, thus multiple imputations were implemented for all regression analyses using a wholly conditional specification with ten iterations, incorporating the admission date as an auxiliary variable. Standard model diagnostics were examined, including goodness of fit, collinearity, and influence.

#### 2.2.3. Sensitivity Analysis

As a sensitivity analysis to remove the effects of mortality from the length of stay, linear regression analysis was used to predict the log-transformed length of hospital stay and to estimate the effect of malnutrition controlling for age, sex, race, body mass index, diabetes, hypertension, diarrhea, COPD, and admission source limited to the sample who survived to hospital discharge. As a sensitivity analysis to account for potential selection bias for malnutrition diagnosis, we assigned weights to each patient based on the inverse of their probability of malnutrition diagnosis, as estimated by propensity scores calculated from a model that included age, sex, race, date of admission, diabetes, hypertension, diarrhea, COPD, admission source, and either the length of stay or mortality status. After assuring a balance between malnutrition groups, regression analyses were run incorporating stabilized weights.

#### 2.2.4. Software

Analyses were performed using SAS 9.4 (SAS Institute Inc., Cary, NC, USA). *p*-values < 0.05 were considered significant in all analyses.

## 3. Results

The characteristics of the COVID-19 inpatient cohort are shown in Table 1. Out of the 4311 patients analyzed for the study, 403 (9.3%) patients were diagnosed with malnutrition compared to the 3908 (90.7%) patients who were not malnourished. The mean age was higher for malnourished patients than for those who were not (68.05 vs. 58.78, respectively; *p* < 0.0001). The median BMI was lower in malnourished patients than those without malnutrition (24.1 vs. 29.0, respectively; *p* < 0.0001). There was no difference in gender between those who were malnourished and those who were not (*p* = 0.213). Whites were more likely to be malnourished than Asian, Black, or other races (*p* < 0.0001). Patients with diabetes, hypertension, diarrhea, or COPD (Chronic Obstructive Pulmonary Disease) were more likely to be malnourished compared to those who were not (*p* = 0.020; *p* < 0.001; *p* < 0.001; *p* = 0.014, respectively). The admission source of malnourished patients was significantly different from those who were not malnourished (*p* < 0.0001). The no malnutrition group’s admission source was 77.4% from home or workplace or non-healthcare facility/court/law enforcement, while in the malnutrition group, only 58.6% were admitted from the same source. (*p* < 0.001). The no malnutrition group’s admission source was 11.2% from a skilled nursing facility, intermediate care facility, or assisted living facility, while in the malnutrition group, 25.8% were admitted from the same source. The mortality rate of malnourished patients was 25.3%, which was significantly higher than the mortality rate of 11.6% in those who were not malnourished (*p* < 0.0001).

The WHO score for malnourished patients was significantly different from the non-malnutrition group (*p* < 0.0001). There were 2956 of 3908 patients (90.4%) in the no malnutrition group who needed ICU care, while 313 of 403 patients (9.6%) required ICU care in the malnutrition group. The median length of hospital stay was longer for the malnutrition group than the no malnutrition group (16.08 days vs. 5.63 days, respectively; *p* < 0.0001).

Table 2 displays the patient characteristics by mortality status, alive or deceased. Of the 4311 patients in the cohort, 3756 (87.1%) survived, while 555 (12.9%) died. The mean age at admission was higher in deceased patients than alive patients (77 vs. 57.42, respectively, *p* < 0.001). The mean BMI of deceased patients (27.66) is lower than the BMI (30.34) in the alive group (*p* < 0.001). Male patients had a higher likelihood of mortality than female patients (*p* = 0.035). Differences in race were associated with differences in the likelihood of mortality (*p* < 0.001). Black patients had a lower likelihood of mortality than non-Black patients (*p* = 0.032). Patients with diabetes, hypertension, diarrhea, COPD, and malnutrition were more likely to have higher mortality (*p* < 0.001). The admission source of the deceased was significantly different from the alive group (*p* < 0.0001). The “alive” patients’ admission source was 78.5% from home or workplace or non-healthcare facility/court/law enforcement, while in the deceased group, only 56.2% were admitted from the same source. The alive patients’ admission source was 9.6% from a skilled nursing facility, intermediate care facility, or assisted living facility, while in the deceased group, 32.6% were admitted from the same source.

To elucidate the effect of malnutrition and other parameters of interest on COVID-19 mortality, a multivariable logistic regression analysis was performed and is reported in Table 3. After adjusting for possible covariates, in the model, patients with malnutrition are 76% more likely to reach mortality than those who are not malnourished (*p* < 0.001). Older patients aged 61–74 years are 198% more likely to die than patients younger than 60 years of age (*p* < 0.0001). Patients older than 75 years of age are 455% more likely to die than patients younger than 60 years of age (*p* < 0.0001). Females are 24% less likely to reach mortality than males (*p* < 0.0065). Patients with COPD are 104% more likely to die than those without COPD (*p* < 0.0001). Patients admitted from a nursing home are 102% more likely to die than those from other admission sources (*p* < 0.0001). Patients’ BMI (*p* = 0.3551) and race (*p* = 0.3871) were not significant to predicting mortality in the model. The comorbidities of diabetes (*p* = 0.2707), hypertension (*p* = 0.0567), and diarrhea (*p* = 0.3619) were not statistically significant in the model. Propensity score analysis found similar results to those patients with malnutrition who were 96% more likely to reach mortality than those who were not malnourished (OR = 1.96, 95% confidence interval 1.74–2.21).

One of the critical metrics contributing to patient mortality is the length of hospital stay. In order to determine which covariates have an impact on the length of hospital stay, a multivariable linear regression predicting the log length of hospital stay in 4293 patients was performed and reported in Table 4. In the model, patients with malnutrition were predicted to have a 105% increase in length of hospital stay (*p* < 0.0001). For every year increase in age, the length of hospital stay is predicted to increase by 1.01% (*p* < 0.0001). Females are predicted to have an 11.4% decrease in hospital length of stay compared to males (*p* = 0.0019). Blacks and other races were predicted to have an 11.3% and 15.6% decrease in hospital length of stay compared to Whites (*p* = 0.0082 and *p* = 0.0024, respectively). Asians’ hospital length of stay was not significantly different from Whites (*p* = 0.0704). Patients with diabetes, hypertension, diarrhea and COPD are predicted to have an increased length of hospital stay by 31.0%, 37.4%, 31.0%, and 75.1%, respectively (*p* < 0.0001). Patients admitted from a nursing home were predicted to have a 23.7% decrease in length of hospital stay compared to those from other admission sources, possibly due to an earlier time to mortality (*p* < 0.0001). Propensity score analysis showed similar results. Patients with malnutrition were predicted to have a 141% increased length of hospital stay (beta = 0.881, 95% confidence interval = 0.80–0.962). Sensitivity analysis limited to those who were discharged alive found similar effects, except that the effect of admission from a nursing home was attenuated and no longer statistically significant (beta = −0.09, *p*-value = 0.212).

Table 5 is a linear regression model predicting the relationship between (log) total hospital length of stay and covariates of interest using a saturated model in the 2931 patients hospitalized in the ICU at some point during their total hospital stay. In the linear regression model, for every 1% increase in BMI, the total hospital length of stay decreased by 0.16% (*p* = 0.0002). Age (*p* = 0.64) and gender (*p* = 0.94) were not associated with a change in total hospital length of stay. Asian (*p* = 0.0029), Black (*p* = 0.0003), or other (*p* < 0.0001) races who went to the ICU were associated with 29.0%, 18.7%, and 33.4% decreased total length of hospital stay, respectively compared to Whites. COVID-19 ICU patients in our cohort with hypertension, diabetes, diarrhea, COPD, and malnutrition were associated with a longer total hospital length of stay (*p* < 0.0001). The length of total hospital stay for ICU patients was predicted to increase by 52.3%, 24.3%, 35.7%, and 60.7% for hypertension, diabetes, diarrhea, and COPD patients, respectively. ICU patients with malnutrition were associated with an 87.9% increase in total hospital length of stay (*p* <0.0001). ICU Patients with a WHO score of Death and Severe were predicted to have a longer hospital length of stay by 54.4% and 127.1%, respectively, compared to ICU patients with a mild/moderate WHO score (*p* < 0.0001). Patients admitted from a skilled nursing facility, intermediate care facility, or assisted living facility were associated with a 29.7% shorter total hospital length of stay compared to patients admitted from the home, workplace, non-healthcare facility, or a court/law enforcement (*p* < 0.0001). Patients from a physician’s office, clinic, or another healthcare facility (*p* = 0.50) were not associated with a change in total hospital length of stay. Patients admitted as transfers from another acute care hospital or ED were not associated with a change in total hospital length of stay (*p* = 0.53).

## 4. Discussion

The results of our study provide unique insights based on the outcomes of 4311 patients with SARS-CoV-2 infection admitted to a tertiary care healthcare system during the initial surge of the COVID-19 pandemic in 2020. In our study, malnutrition was confirmed in 9.3% (403/4311) of COVID-19 adult inpatients. As expected, there was a higher age (68.05 vs. 58.78, respectively; *p* < 0.0001) and lower BMI (24.1 vs. 29.0, respectively; *p* < 0.0001) in COVID-19 adult inpatients with malnutrition (*p* < 0.0001). Consistent with our prior study of malnutrition characteristics in an adult medical ward [11], we report no difference in the prevalence of malnutrition by gender in 4311 COVID-19 adult inpatients. COVID-19 adult inpatients who were malnourished were more likely to have diabetes, hypertension, diarrhea, or COPD, compared to those who were not malnourished (*p* = 0.020; *p* < 0.001; *p* < 0.001; *p* = 0.014, respectively). While hypertension is a known adverse prognostic risk factor for COVID-19 outcomes, its association as a malnutrition risk factor has not been previously described [16]. COPD is a disease associated with a higher risk of sarcopenia and malnutrition due to the adverse consequences of cor pulmonale, causing higher energy demands to maintain respiratory effort and right-heart failure causing protein-losing enteropathy and nutrient malabsorption [17,18]. Diarrhea has been reported to be associated with an adverse prognosis in a subset of COVID-19 patients with multiple purported mechanisms, including gut microbial dysbiosis, malabsorption, leaky gut, proinflammatory cytokines, protein-losing enteropathy, and malabsorption of immune-modulating nutrients, such as 25-hydroxyvitamin D [19,20]. The admission source of malnourished patients was significantly different from those who were not malnourished (*p* < 0.0001). We observed that 104/541 (19.2%) of SARS-CoV-2 infected adults admitted from a skilled nursing facility, intermediate care facility, or assisted living facility had malnutrition. Furthermore, COVID-19 inpatients with malnutrition were more likely to have been admitted from a skilled nursing facility, intermediate care facility, or assisted living facility, when compared to those without malnutrition (25.81% vs. 11.2%, respectively). These findings are consonant with data reported by Faxen-Irving et al. that the prevalence of malnutrition in a nursing home setting was 17.2% [21].

In the present study, we observed that the prevalence of malnutrition varies by race. In COVID-19 adult inpatients, Whites were more likely to be malnourished than Asian, Black, or other races (*p* < 0.0001). We previously explored the issue of possible racial and gender disparities in ordering oral nutrition supplements (ONSs) for adult inpatients [22]. In our previous study of 8711 adult inpatients with malnutrition, multivariate regression analyses showed that African Americans (*p* = 0.014) and females (*p* < 0.001) were less likely to receive ONS when compared to their racial and gender counterparts [22]. Likewise, an earlier study of parenteral nutrition (PN) ordering for Canadian adult inpatients with inflammatory bowel disease and malnutrition also showed that African Americans were less likely to receive inpatient PN than Whites (odds ratio (OR) 0.67; 95% confidence (CI), 0.5–0.89) [23]. Consonant with these prior studies, interventions for adult inpatients with other disease entities have also shown concerning the trends in racial disparities [24,25]. Large urban community settings with high poverty rates are prone to malnutrition, primarily affecting African Americans [26]. For instance, a sample of 10,001 ethnically diverse community-dwelling adults aged 55 years plus in Chicago, Illinois, showed that Blacks were more likely to be at nutritional risk than Whites [27].

Our reported prevalence of 9.3% malnutrition in 4311 consecutive adult inpatients with COVID-19 varies from limited reports in the literature. Allard et al. observed in a small French cohort of 108 COVID-19 adult inpatients that 38.9% had malnutrition based on regional guidelines (BMI of 18.5 or less or having a history of weight loss of 5% within 3 months or 10% within 6 months) [28]. They did not associate malnutrition with COVID-19 disease severity or outcomes. Da Porto et al. reported a malnutrition prevalence of 24.3% in 150 consecutive adult patients admitted for COVID-19 pneumonia in Spain using bioelectrical impedance velocity analysis [29]. They noted during the 60 days of follow-up, 10/37 (27%) were malnourished patients, and 13/108 (12%) non-malnourished patients required invasive mechanical ventilation (*p* = 0.023). However, we observed no differences in the use of oxygen, hi-flow oxygen, or the need for mechanical ventilation regardless of malnutrition status. Bedock et al. used the Global Leadership Initiative for Malnutrition (GLIM) criteria to diagnose malnutrition in 48 of 114 patients admitted to a medical ward with COVID-19 [30]. Li et al. reported the prevalence of malnutrition to be 52.1% in a cross-sectional study of 182 COVID-19 inpatients older than 65 years of age in Wuhan, China, using the Mini Nutritional Assessment (MNA) criteria [31]. Relative to these prior studies, we report a lower prevalence of malnutrition in COVID-19 inpatients in 2020 (9.3%), though higher than our pre-COVID-19 institutional malnutrition prevalence of 3.1% [22]. Possible reasons for the discrepancy in malnutrition prevalence for COVID-19 inpatients across these studies include variability in the severity of illness, age, comorbidities, source of admission, and the methodologies utilized to establish the malnutrition diagnosis, which presently lacks global uniform standards [12,32,33,34,35].

Malnutrition has been well-described as an independent predictor of adverse health-economic outcomes, such as length of stay and mortality [28,36,37]. Overall, 3756 of 4311 (87.1%) patients in the cohort survived, while 555 (12.9%) died. The mortality rate in our study was higher in malnourished patients (102/403; 25.31%) than those who were not malnourished (453/3908; 11.59%, *p* < 0.0001). Likewise, the WHO score, which prognosticated COVID-19 disease severity, was higher in the malnourished than in the no-malnutrition group (*p* < 0.0001). To isolate the effect of malnutrition from other confounding parameters of interest in COVID-19 mortality, the multivariable logistic regression analysis revealed that adult inpatients with malnutrition were 76% more likely to reach mortality than those who were not malnourished (*p* < 0.001). In our cohort, male patients had a higher likelihood of mortality than females (*p* = 0.035). Differences in race were associated with differences in the probability of mortality (*p* < 0.001). Surprisingly, Black patients had a lower likelihood of mortality than non-Black patients (*p* = 0.032). Other independent predictors of higher mortality in our cohort include an age greater than 60 years, admission source from a nursing facility, and COPD. In this model, females were 24% less likely to die than their male counterparts.

There is a well-known relationship between higher BMIs and increased mortality of COVID 19-patients in the critical care setting [38]. However, lower BMI appears to be an important variate of mortality in COVID-19. Da Porto et al. conducted a mortality analysis based on malnutrition status in a smaller cohort of 150 adult inpatients with COVID-19 pneumonia [29]. Similar to the findings that we report in this paper, they observed a higher mortality rate (13/37; 35%) in COVID-19 inpatients with malnutrition when compared to non-malnourished patients (9/113; 8%) (*p* < 0.001) [29]. Bedock et al. reported a trend for a higher risk of mortality in 114 COVID-19 patients with weight loss above 5% of initial weight (OR: 3.7, 95% CI 1.0; 26.5, *p* = 0.09) [30]. However, malnutrition was not associated with the risk of transfer to ICU or death. Based on our data, and consistent with others, patients infected with COVID-19 who are at nutritional risk appear to have more adverse outcomes. Di Filippo et al. conducted nutrition risk screening of 213 COVID-19 patients at the Emergency Department of San Raffaele University Hospital using the mini nutritional assessment (MNA) tool [39]. They noted that 28.6% (61/213) of these patients presented with a loss of more than 5% of initial body weight. These patients had greater systemic inflammation and longer disease duration than those who did not lose weight at presentation [39]. We also report that the BMI of deceased COVID-19 inpatients is lower than those who survive (27.6 vs. 30.3, respectively; *p* < 0.001), but is not a predictor of mortality (OR 0.99; 95% CI 0.98–1.01; *p* = 0.35). Elderly COVID-19 inpatients older than the age of 65 years with moderate–severe malnutrition diagnosed by the Geriatric Nutritional Risk Index (GNRI), were shown to be associated with a higher risk of mortality (HR 9.285 [1.183–72.879], *p*  =  0.034) [40]. Overall, malnutrition is an independent risk factor for mortality in COVID-19, and its prevalence and impact appear to be more pronounced in the elderly.

Hospital length of stay (LOS) is a health-economic outcome metric linked to malnutrition [11,41]. In our study, the median length of hospital stay was significantly longer for the malnutrition group than the no malnutrition group (16.08 days vs. 5.63 days, respectively; *p* < 0.0001). Interestingly, differences in race (*p* = 0.001) were associated with differences in the length of stay as White patients are predicted to have a 15% increase in length of hospital stay compared to other races (*p* = 0.0008). The female gender was predictive of an 11.4% decrease in hospital length of stay compared to males (*p* = 0.002). The improved health economic outcomes of females compared to males admitted for COVID-19 may relate, in part, to known sex-based differences in immunity [42]. Aside from malnutrition and race, other factors also may influence hospital length of stay in our cohort. Patients with diabetes, hypertension, diarrhea, and COPD, were more likely to have a longer hospital length of stay (*p* < 0.001) by 31.0%, 39.1%, 31.0%, and 75.1%, respectively. Differences in the source of admission to the hospital affected the length of stay. However, the findings were paradoxical, as patients admitted from the nursing home were predicted to have a 23.7% decrease in length of hospital stay compared to those from other admission sources (*p* < 0.0001). Overall, 9.3% (403/4311) of adult COVID-19 patients were diagnosed with malnutrition and were predicted to have a 105% increase in length of hospital stay compared to those without malnutrition (*p* < 0.0001).

The intensive care unit (ICU) COVID-19 patients present challenges to meet their caloric needs due to proning, intolerance to gastrointestinal feedings, gastrointestinal dysmotility, gut microbial dysbiosis, and high energy demands. We analyzed the data from 2931 ICU COVID-19 patients in our cohort. For every 1% increase in BMI, the length of stay decreased by 0.16% (*p* = 0.0002), which supports prior studies linking sarcopenia to adverse outcomes in critically ill patients [43,44].

Surprisingly, being Asian (*p* = 0.0029), Black (*p* = 0.0003), or other non-White race (*p* < 0.0001) were associated with decreased ICU length of stay compared to Whites. Dotson et al. previously reported that among 4377 children hospitalized for Crohn’s disease, Black children were associated with more extended hospital stays when compared to Whites (*p* < 0.001) [45]. To our knowledge, this is the first study reporting a racial disparity in-hospital stay in the ICU setting. Linear regression models in our COVID-19 cohort showed that hypertension, diabetes, diarrhea, COPD, and malnutrition were independent predictors of prolonged ICU stay (*p* < 0.001). Overall, ICU patients with COVID-19 and malnutrition in our cohort were associated with an 87.9% increased length of stay (*p* <0.0001).

During the COVID-19 pandemic, individuals infected with symptomatic SARS-CoV-2 infection may experience nutritional compromise, impaired immunity and their ability to mitigate the viral illness [10,46]. The hypermetabolic response to SARS-CoV-2 infection can include fever, tachycardia, tachypnea, and cytokine storm that increase resting energy expenditure, caloric and micronutrient requirements. Gastrointestinal manifestations, such as diarrhea from dysbiotic gut microbial shifts, anorexia, coupled with loss of taste, are frequent causes of anorexia and curtailed nutrient intake and even promote malabsorption [47]. Pre-existing sarcopenia and malnutrition, as is often observed in the elderly, render them susceptible to further nutritional compromise from food insecurity, social isolation, depression, and loss of lean body mass from confinement [48]. The catabolic state is evident upon admission with sarcopenia and elevated inflammatory markers with the depletion of visceral proteins leading to protein-calorie malnutrition and immunocompromise, which collectively impose a high risk for viral virulence and adverse outcomes [46]. In our study, being 61 years old or older with malnutrition was an independent predictor for COVID-19 mortality.

Oral nutrition supplementation in malnourished elderly adults improves immunocompetence [49]. Once hospitalized, early screening and intervention of malnutrition with ONS have been previously shown to improve hospital length of stay and 90-day mortality [12,13,50]. Moran-Lopez et al. conducted a retrospective cohort chart review of 75 hospitalized patients with COVID-19 to ascertain if early feeding improved outcomes [51]. Nutritional support was provided to only 21 of 75 COVID-19 adult inpatients (28%) and was started on early enteral nutrition (EEN) in only 12 patients (16%). In those who received EEN, hospital length of stay was shorter, respiratory distress was less frequent and severe, and fewer complications were observed in this small cohort. At present, there is a dearth of controlled oral nutritional supplement or early feeding trials for COVID-19 [52]. Since malnutrition appears to be an independent risk factor and predictor for prolonged hospital length of stay and mortality [53], efforts to facilitate early enteral dietary management engagement (FEEDME) [11] should be considered while adhering to rigorous scientific principles [54].

As with most studies, our study has limitations to consider in the interpretation of the findings. Although our sample size was reasonable, the prevalence of malnutrition was insufficient to allow for analysis within the subset of malnourished patients. Known predictors of COVID-19 severity, including inflammatory measures, were not available at the time of analysis. While our data came from a single health system, this system included five hospitals in a large geographic area; nonetheless, these results may not be generalizable outside of the Baltimore–Washington area. Finally, as with all observational research, there is a potential for confounding due to unmeasured effects.

## 5. Conclusions

Malnutrition is an independent predictor for a longer hospital stay, increased ICU length of stay, and higher hospital mortality from COVID-19 illness. Malnutrition is highly prevalent in elderly individuals and continues to be an important predictor of mortality in aged populations hospitalized with COVID-19. In our study, unique racial and gender disparities in malnutrition prevalence and health-economic outcomes in the hospital setting were observed and deserve further validation. Interventional trials of early screening and high-protein oral nutritional supplementation should be considered in sarcopenic individuals admitted to the hospital for COVID-19 illness.

## Figures and Tables

**Table 1 nutrients-14-01310-t001:** Sample characteristics and outcomes by malnutrition status.

	No Malnutrition n = 3908; 90.7%	Malnutrition n = 403; 9.3%	
**Characteristic**	**Mean (SD)**	**Mean (SD)**	***T*-Test *p*-Value**
Age (years)	58.78 (19.3)	68.05 (17.2)	<0.0001
**Characteristic**	**Median (IQR)**	**Median (IQR)**	**Wilcoxon Test *p*-Value**
Body mass index (kg/m^2^) *	29.0 (25.0–34.3)	24.1 (20.2–29.9)	<0.0001
**Characteristic**	**n (%)**	**n (%)**	**Chi-Squared Test *p*-Value**
Male gender	1909 (48.9)	210 (52.1)	0.213
Race			<0.0001
Asian	184 (4.7)	13 (3.2)	
Black	1435 (36.7)	140 (34.7)	
Other	936 (24.0)	49 (12.2)	
White	1353 (34.6)	201 (49.9)	
Comorbidity			
Diabetes	1618 (41.4)	191 (47.4)	0.020
Hypertension	2639 (67.53)	341 (84.6)	<0.0001
Diarrhea	759 (19.42)	116 (28.8)	<0.0001
COPD	308 (7.9)	46 (11.4)	0.014
Admission Source	3022 (77.4)	236 (58.6)	<0.0001
Home or workplace or non-health care facility; court/law enforcement			
Physician’s office or clinic; other health care facility	249 (6.4)	27 (6.7)	
Skilled nursing facility, intermediate care facility or assisted living facility	437 (11.2)	104 (25.8)	
Transfers from another acute care hospital or ED	195 (5)	36 (8.9)	
Outcomes			
Mortality	453 (11.6)	102 (25.3)	<0.0001
ICU Patients	N (%); N = 2956; 90.4%	N (%); N = 313; 9.6%	Chi-Squared Test *p*-Value
WHO Score **			<0.0001
3 (hospitalized, no oxygen therapy)	561 (19.0)	38 (12.1)	
4 (oxygen)	1389 (47.0)	106 (33.9)	
5 (NIPPV or hi-flow oxygen)	264 (8.9)	16 (5.1)	
6 (intubation and mechanical ventilation)	66 (2.2)	5 (1.6)	
7 (ventilation and additional organ support)	289 (9.8)	73 (23.3)	
8 (death)	387 (13.1)	75 (24.0)	
	Median (IQR)	Median (IQR)	Wilcoxon Test *p*-Value
Length of hospital stay (days)	5.63 (2.73–13.19)	16.08 (6.32–56.73)	<0.0001

* N with BMI data in no malnutrition group = 3555; N with BMI data in malnutrition group = 375; ** WHO score available only for those in ICU, N in no malnutrition group = 2956, N in malnutrition group = 313.

**Table 2 nutrients-14-01310-t002:** Characteristics of COVID-19 inpatients based on mortality status, alive or deceased, who were admitted to a JHH affiliated hospital between 1 March 2020 and 3 December 2020 (n = 4311).

	Alive	Deceased		
	n = 3756; 87.1%	n = 555; 12.9%		
Variables	Mean (SD)	Mean (SD)	Statistical Test	*p*-Value
Age (years)	57.42 (18.9)	74.76 (14.3)	*T*-test	<0.0001
Body mass index (kg/m^2^) *	30.34 (8.2)	27.66 (7.9)	Wilcoxon	<0.0001
	**n (%)**	**n (%)**		
Gender				
Female	1933 (51.5)	259 (46.7)	Chi-square	0.034834
Male	1823 (48.5)	296 (53.3)		
Race				
Asian	169 (4.5)	28 (5.1)	Chi-square	<0.0001
Black	1395 (37.1)	180 (32.4)		
Other	917 (24.4)	68 (12.3)		
White	1275 (34.0)	279 (50.3)		
Black				
No	2361 (62.9)	375 (67.6)	Chi-square	0.031551
Yes	1395 (37.1)	180 (32.4)		
Comorbidity				
Diabetes	1537 (40.9)	272 (49.0)	Chi-square	0.000314
Hypertension	2501 (66.6)	479 (86.6)	Chi-square	<0.0001
Diarrhea	766 (20.4)	109 (19.6)	Chi-square	0.680012
COPD	249 (6.6)	105 (18.9)	Chi-square	<0.0001
Malnutrition	301 (8.0)	102 (18.4)	Chi-square	<0.0001
WHO Score (Grouped) **				
Death	0	462 (93.7)	NA	
Mild/moderate	2077 (74.8)	17 (3.5)		
Severe	699 ((25.2)	14 (2.8)		
Admission Source (Grouped) ***				
Home or workplace or non-health care facility; court/law enforcement	2946 (78.5)	312 (56.2)	Chi-square	<0.0001
Physician’s office or clinic; other health care facility	245 (6.5)	31 (5.6)		
Skilled nursing facility, intermediate care facility or assisted living facility	360 (9.6)	181 (32.6)		
Transfers from another acute care hospital or ED	200 (5.3)	31 (5.6)		

* N with BMI data in alive group = 3460; N with BMI data in deceased group = 470; ** WHO score available only for those in ICU, N in alive group = 2776, N in deceased group = 493; *** N with Admission Source data in alive group = 3751; N in deceased group = 555.

**Table 3 nutrients-14-01310-t003:** Multivariable logistic regression analysis predicting mortality (n = 4311).

Parameter	Odds Ratio	95% Confidence Interval	*p*-Value
Malnutrition	1.76	1.34–2.30	<0.0001
Age 61–74 (vs. age ≤ 60; years)	2.98	2.22–3.99	<0.0001
Age ≥ 75 (vs. age ≤ 60; years)	5.55	4.07–7.57	<0.0001
Body mass index (kg/m^2^)	0.99	0.98–1.01	0.3551
Female gender	0.76	0.63–0.93	0.0065
White race	1.09	0.89–1.34	0.3871
Diabetes	1.12	0.91–1.38	0.2707
Hypertension	1.33	0.99–1.77	0.0567
Diarrhea	0.89	0.70–1.14	0.3619
COPD	2.04	1.56–2.66	<0.0001
Admission from nursing home	2.02	1.61–2.55	<0.0001

**Table 4 nutrients-14-01310-t004:** Multivariable linear regression predicting (log) length of hospital stay (n = 4293).

Parameter	Beta Estimate	95% Confidence Interval	*p*-Value
Malnutrition	0.72	0.587–0.854	<0.0001
Age (years)	0.01	0.004–0.009	<0.0001
(log) Body mass index (kg/m^2^)	−0.08	−0.247–0.088	0.3536
Female gender	−0.12	−0.197–−0.045	0.0019
Black race	−0.12	−0.216–−0.032	0.0082
Asian race	−0.17	−0.363–0.014	0.0704
Other race	−0.17	−0.277–−0.06	0.0024
White race	reference		
Diabetes	0.27	0.188–0.354	<0.0001
Hypertension	0.32	0.218–0.417	<0.0001
Diarrhea	0.27	0.174–0.365	<0.0001
COPD	0.56	0.416–0.698	<0.0001
Admission from nursing home	−0.27	−0.397–−0.149	<0.0001

**Table 5 nutrients-14-01310-t005:** Multivariable linear regression predicting the (log) hospital length of stay, ICU patients only (n = 2931).

Parameters		Estimate	Standard Error	Wald Chi-Square	Pr > ChiSq
Intercept		3.1011	0.3822	65.84	<0.0001
Age (years)		0.0008	0.0017	0.22	0.6425
(log) Body mass index (kg/m^2^)		−0.3779	0.1017	13.81	0.0002
Male		0.0036	0.0472	0.01	0.9391
Female		reference	0		
Race					
Asian		−0.342	0.1147	8.9	0.0029
Black		−0.2073	0.057	13.24	0.0003
Other		−0.4063	0.0674	36.4	<0.0001
White		reference	0		
Comorbidity					
Hypertension	Yes	0.4205	0.0615	46.76	<0.0001
	No	reference	0		
Diabetes	Yes	0.2177	0.0511	18.12	<0.0001
	No	reference	0		
Diarrhea	Yes	0.3053	0.0574	28.25	<0.0001
	No	reference	0		
COPD	Yes	0.4743	0.0828	32.83	<0.0001
	No	reference	0		
Malnutrition	Yes	0.6305	0.0811	60.41	<0.0001
	No	reference	0		
WHO Score Grouped					
Death		0.4346	0.0742	34.29	<0.0001
Severe		0.8204	0.0583	197.74	<0.0001
Mild/moderate		reference	0		
Admission Source Grouped					
Physician’s office or clinic; other health care facility		0.0638	0.0955	0.45	0.5045
Skilled nursing facility, intermediate care facility or assisted living facility		−0.353	0.0787	20.14	<0.0001
Transfers from another acute care hospital or ED		0.0631	0.0999	0.4	0.5277
Home or workplace or non-health care facility; court/law enforcement		reference	0		

## Data Availability

The data utilized for this publication were part of the JH-CROWN: The COVID-19 PMAP Registry [55], which is based on the contribution of many patients and clinicians. De-identified data may be made available through a data use agreement and an approved Johns Hopkins Medicine Institutional Review Board protocol (available online: https://www.hopkinsmedicine.org/institutional_review_board/news/covid19_information/index.html (accessed on 22 February 2022). Code for analyses is available from the corresponding author upon request.

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
