# Peer review of "Malnutrition Increases Hospital Length of Stay and Mortality among Adult Inpatients with COVID-19"

_nutrients, 2022, doi:10.3390/nu14061310_

Round 1
Reviewer 1 Report
I congratulate the authors on the work done on this relevant subject.
Malnutrition in critical care is highly prevalent and well documented to have adverse implications on morbidity and mortality. However, nutrition is an important element of care. The nutritional assessment and the early nutritional care management of COVID-19 patients must be integrated into the overall therapeutic strategy.
I would like to contribute with some comments, suggestions and questions to the authors:
- In abstract: please delete numbers e.g. (1) Background etc.
- I could not find an explanation of the factors that are needed to calculate the MUST score. The authors should add a paragraph in the methods section were they outline the score.
- Discussion: 369- it is worth add that in Czapla et al. study being overweight in critically ill COVID-19 patients requiring invasive mechanical ventilation increases their risk of death significantly https://www.mdpi.com/2072-6643/13/10/3302/htm
- Please add study limitation on the end of discussion or like a separate point before conclusion section.
Overall, the authors have analyzed the available data to a reasonable conclusion
Author Response
We deleted the numbers in the abstract per your recommendations. The numbers were pre-populated with the Nutrients Microsoft Word template.
We revised and included the citation for the MUST score.
Weaved in the Czapla et al. study-see line 378-379.
Move the limitations to the final paragraph per your recommendations.
Reviewer 2 Report
This study presents a well-executed epidemiological design (retrospective, observational study; IRB00253531) aimed at evidencing the proximal and distal determinants associated with concurrent malnutrition (crude prevalence 9.3%) and its effect on the clinical evolution and prognosis of COVID-19 inpatients admitted to a Johns Hopkins-affiliated hospital. Age, BMI, ethnicity, and chronic comorbidities differed between undernourished and well-nourished patients; higher mortality and Ventilation and Additional Organ Support need and lengthen of hospital stay were associated with undernourishment. Undernourished patients were 76% more likely to reach mortality than those well-nourished. Mortality predictors were Age, diagnosed COPD, admission from a nursing home, malnutrition. Minor changes may improve the scientific soundness of the study:
- Tables. Please format all tables according to nutrients´ guidelines.
- Results & discussion. The calculation of other epidemiological indicators such as the "number needed to treat (NNT)" can reinforce the idea of the need to treat malnutrition clinically, despite other non-modifiable covariant factors (e.g. age, ethnicity). Please restrict any comments or hypotheses not supported by the study findings (e.g. need for oral nutritional supplementation, presence of immunosuppression, vitamin D deficiency) or relocate them in a paragraph at the end as "recommendations". Also, add a paragraph where the weaknesses and strengths of the study are declared, as well as future projections.
- References. They are too many, though very recent. Performing the suggested modifications to the discussion section may help to reduce the number of unnecessary references.
Author Response
Thank you for your insightful feedback. Regarding the formatting of tables, we used the Microsoft Word document template provided by Nutrients including the pre-formatted tables. We asked the editorial office to investigate and advise.
We removed a paragraph on the oral nutrition supplements, vitamin D, etc as you have advised.
Reviewer 3 Report
Thank you for the opportunity to review the paper “Malnutrition Increases Hospital Length of Stay and Mortality 2 Among Adult Inpatients with COVID-19” by Tyrus Vong et al.
The authors investigated malnutrition, using validated screening tools (MUST), in a wide and heterogeneous and large cohort of patients.
The authors concluded that malnutrition is an independent predictor for longer hospital and intensive care unit stay and higher mortality in COVID patients and that further trials on early detection and treatment of this condition - with ONS, for example - are desirable.
Researchers conducted good quality work, the writing is clear and the study method appropriate. It is of conspicuous interest to readers.
I have only one major concern:
The Authors stated that "Malnourishment was identified using the malnutrition universal screening tool (MUST)". There is a matter of definitions.
The MUST is a validated screening tool useful for identifying patients "at risk of malnutrition" NOT necessarily malnourished. It is an easy to perform method applicable by any healthcare professional; however, the diagnosis of malnutrition needs to be done by specialized personnel according to GLIM Criteria (doi: 10.1016/j.clnu.2018.08.002). So, it is improper in a Nutrition Journal to describe MUST positive patients as malnourished. I suggest to revise the paper according to GLIM Criteria or, alternatively changing the cohorts ad "Not at risk" and "At risk of malnutrition", revising the definition throughout the paper.
Minor remark:
- Line 78: Replace “consists” with “consist”
Author Response
Thank you for your insightful comments. The malnutrition universal screening tool (MUST) is a validated screening tool for nutrition risk screening performed by the registered nurses but was not used for confirmation of malnutrition. In the manuscript, we noted that malnutrition was "confirmed by registered dietitians "but didn't specify the guidelines used. We did use GLIM but failed to specify and reference it (since it is our institutional guideline). Good pickup. We also replaced "consists" with "consist". Thank you
Round 2
Reviewer 1 Report
It's ready for publication
Author Response
Thank you for your comments.
Reviewer 3 Report
Ok, the authors addressed my concerns.
I have no other questions.
Best.
Author Response
Thank you for your comments.